# Artificial Neural Network (ANN)-Based Pattern Recognition Approach Illustrates a Biphasic Behavioral Effect of Ethanol in Zebrafish: A High-Throughput Method for Animal Locomotor Analysis

**DOI:** 10.3390/biomedicines11123215

**Published:** 2023-12-04

**Authors:** Vladislav O. Myrov, Aleksandr I. Polovian, Sofiia Kolchanova, Georgii K. Galumov, Helgi B. Schiöth, Dmitrii V. Bozhko

**Affiliations:** 1ZebraML, Inc., Houston, TX 77043, USA; 2Department of Surgical Sciences, Division of Functional Pharmacology and Neuroscience, Uppsala University, 751 24 Uppsala, Sweden; helgi.schioth@neuro.uu.se

**Keywords:** zebrafish, ethanol, time-dependent analyses, artificial neural networks

## Abstract

Variations in stress responses between individuals are linked to factors ranging from stress coping styles to the sensitivity of neurotransmitter systems. Many anxiolytic compounds can increase stressor engagement through the modulation of neurotransmitter systems and are used to investigate stress response mechanisms. The effect of such modulation may vary in time depending on concentration or environment, but those effects are hard to dissect because of the slow transition. We investigated the temporal effect of ethanol and found that ethanol-treated individual zebrafish larvae showed altered behavior that is different between drug concentrations and decreases with time. We used an artificial neural network approach with a time-dependent method for analyzing long (90 min) experiments on zebrafish larvae and found that individuals from the 0.5% group begin to show locomotor activity corresponding to the control group starting from the 60th minute. The locomotor activity of individuals from the 2% group after the 80th minute is classified as the activity of individuals from the 1.5% group. Our method shows three clusters of different concentrations in comparison with two clusters, which were obtained with the usage of a statistical approach for analyzing just the speed of fish movements. In addition, we show that such changes are not explained by basic behavior statistics such as speed and are caused by shifts in locomotion patterns.

## 1. Introduction

Zebrafish are vertebrates with complex brains and behaviors and are increasingly gaining interest as a convenient model system for drug screenings because they are small with a relatively short life cycle and are much easier and less expensive to house in the lab compared with rodents [1,2,3]. This model system enables researchers to combine complex behavioral phenotyping with high-throughput chemical screening. Zebrafish possess orthologs of about 70% of human disease-associated genes [4,5] and exhibit a broad range of disease-associated phenotypes, including disorders of physiology, metabolism, and behavior.

The usage of zebrafish models is growing in various types of research, from cancer [6] and aging [7] to the role of GABAergic signalling in motor diseases [8], mechanisms of anti-epileptic drugs [9,10], and the effect of neuroactive drugs such as ethanol [11] and LSD [12]. Zebrafish screening experiments have facilitated the discovery of many compounds that have eventually made it to clinical trials, including new compound classes and repurposed drugs [13]. In addition, accelerating advances in genome editing and high-throughput phenotyping have the potential to bring about even more applications for zebrafish in drug discovery in the coming years, including the discovery of compounds that mitigate human disease phenotypes associated with specific genetic variants [13,14].

Trained human observers commonly use a behavior classifier to detect various kinds of changes in motor activity patterns in fish and distinguish between them [15]. The observers record and subsequently interpret all the details of motor activity in a particular experiment. One of the most critical limitations of this technique is its extreme labor intensiveness, as large volumes of video need to be visually analyzed. The amount of work that is needed to go through each of the recordings manually can have negative impacts on the accuracy and consistency of the interpretations. The duration of video recordings in such human-observed settings usually does not exceed 10 min per fish, and, therefore, high-throughput assays are not very feasible.

There is increasing interest in specialized software capable of overcoming these shortcomings [16,17]. Such approaches aim to enable longer experiment recordings, increasing the capacity to recognize the specific motor activity of the fish, such as being able to calculate a set of movement macro-parameters (such as speed, time spent in different areas of the aquarium, duration of freezing behavior), which are later subjected to statistical analysis [18]. This software can single out the set of parameters that demonstrate statistical significance. Another existing approach uses Motion Index calculation [19], describing fish movements in the aquarium and allowing for building an activity profile over the entire experiment duration. However, the major limitation of the analyses based solely on macro-parameters is the lack of testing for alterations in the fish locomotor track. For instance, it does not allow one to differentiate between chaotic and circular movements, which a human observer can easily notice.

Data analysis with machine learning (ML) can combine the methods mentioned in the previous paragraph. ML approaches that lie in the field of unsupervised learning (hierarchical clustering, PCA, random forests) show high potential in the area of medical research. Their efficiency was demonstrated, for example, in a study of brain activity maps by [20]. ANN (artificial neural network)-based machine learning methods are also used to predict the properties of tested substances and their targets [16,21] for accelerated diagnostics (using pattern recognition) in diseases such as cancer and diabetic retinopathy [17,22], as well as for processing and combining clinical and genomic data [23,24]. Statistical descriptors are widely used and easy to understand [25,26], but they have several limitations [27]. First, one has to design them manually; therefore it is impossible to include all possible traits. Another problem is that statistical descriptors cannot solve an inverse task: predict the most probable drug affecting a fish, observe activity statistics.

Here, we aim to overcome these limitations by training an ANN using movement tracks as features and drugs as training targets. We applied a previously established ANN-based pattern recognition approach to the problem of fish locomotor activity pattern classification [27]. We extended it to analyze the time-dependent alterations in such activity. This project may contribute to increasing the speed and throughput of preclinical research trials, improving the precision of such tests, and mitigating human interpretation errors.

## 2. Materials and Methods

In this study, we designed and performed two independent in vivo experiments with the same setup including animals, data collection, and processing. This approach helps us to reduce “experimenter bias”.

### 2.1. Animals and Housing

Wild-type zebrafish of an undefined strain and approximately 3 months of age were obtained from a local pet shop. After purchase, the fish were subjected to a 30-day quarantine, after which they were transferred to system tanks. Husbandry and housing were conducted according to the international standards of zebrafish care [28]. Briefly: adult fish were kept in a flow-through recirculation system under standard conditions: a light/dark cycle of 14/10 h and a temperature of 28.5 °C. They were fed twice daily with Tetramin dry flakes (Tetra GmbH, Melle, Germany) and Artemia salina nauplii. Fish aged 6–8 months were used for breeding. Eggs were obtained as a result of natural spawning. Two hours post spawning, embryos were sorted under a Nikon SMZ 1500 stereomicroscope (Nikon, Tokyo, Japan) and unfertilized eggs were removed. Normal embryos were placed in Petri dishes and cultured in egg water at a temperature of 28.5 °C until the start of active feeding. Then, the larvae were placed in 250 mL glasses, with 25 larvae per 200 mL of egg water (60 μg of sea salt in 1 mL). Larvae were kept at 28.5 °C, with a day/night cycle change of 14/10 h, and fed with paramecia.

For the experiments, larvae aged 11–13 dpf (days post fertilization) were selected (see Table 1 for the exact age). This age was chosen to ensure completely developed and functioning sensory and motor systems, allowing for studies of locomotor behavior [29], and to fulfill the size requirements of the detection software.

### 2.2. Ethical Statement

Ethical principles for use of laboratory animals were followed in accordance with the European Convention for the Protection of Vertebral Animals Used for Experimental and Other Scientific Purposes, (ETS No. 123). The study protocol was reviewed and approved by N.N. Petrov NMRC of Oncology local ethics committee (Protocol No. 8 of 18 June 2020).

### 2.3. Data Collection

Ten minutes before the start of the video recording, the larvae were placed into the wells of a 24-well plate, a single larva per well, with a minimum amount of medium, using a Pasteur pipette. Ethanol was used for both in vivo experiments (Ethyl alcohol, pure, Product No. 493511, CAS No.: 64-17-5, MERCK, Oakville, ON, Canada). Ethanol (12 wells) or egg water (another 12 wells) solutions (see Table 2) were added to the wells. The volume of the medium in each well was 0.5 mL. The video recording was conducted at a room temperature of 24 °C.

A network camera with GigE Blackfly BFLY-PGE-13H2M (FLIR, Willsonville, ON, Canada) interface was used. The video was recorded to ReadyNAS 2120 (Netgear, San Jose, CA, USA) with a RAID5 array of 4 4Tb WD40PURX surveillance (Western Digital, San Jose, CA, USA). We used the M-JPEG video format with the Point Grey FlyCapture2 software (v2.13.3.61, FLIR, Willsonville, ON, Canada).

We recorded all experiments with the following video properties: 1250 × 850 pixel resolution, at least 90 min of recording duration, and 25 frames per second. Figure 1 shows how both in vivo experiments were designed.

### 2.4. Data Pre-Processing

We followed the pre-processing pipeline described in [27]. The procedure was the same for every animal in the dataset. Fish coordinates were obtained by recording the well plate in horizontal projection and extracted with the idTracker software v2.1 [30]. A total of 12 fish were recorded for each group.

The 90 min of coordinates are clipped into a sequence of 30 s consecutive coordinate pairs (X, Y). This pre-processing algorithm generates 180 sequences per fish and 2160 sequences per ethanol/control group in each experiment. The total control sequences are 17,280 in each experiment.

### 2.5. Neural Networks

We used a Convolution Neural Network (CNN) architecture with ResNet50 as a backbone [31] to classify the chemical substance or its concentration that fish were exposed to. To train the network, we used a graphical representation of fish locomotion. We cut each movement track into frames of equal length (time = 30 s) and drew it as an image using an XZ projection. Each track segment (a line between two points) was colored according to the relative speed of the corresponding segment.

To optimize model parameters, Adam optimizer [32] was used (learning rate 0.001, momentum 0.9) alongside the Focal loss [33] function. Focal loss is defined as
(1)$$FocalLoss\left(p_{t}\right)=-1{\left(1-p_{t}\right)}^{\gamma }log\left(p_{t}\right),$$
where γ≥0 is a focusing parameter and $p_{t}$ is a vector of prediction probabilities and allows one to reduce the effect of unbalanced classes presented due to the number of control recordings.

### 2.6. In Silico Experiments

To classify the recorded video tracks, we performed a multi-class in silico experiment. We trained a neural network to predict an active compound given a short segment of a locomotor track. To train the network, we used recordings of fish behavior after inducing different concentrations of ethanol, where each ethanol concentration (class) had its label, and all controls were merged into a single class. The training procedure was made on data from the first in vivo experiment, and for the test procedure, we used data from the second independent in vivo.

To assess the quality of model predictions, we computed accuracy across classes—that is, the fraction of correct predictions. We built a confusion matrix of size MxN, where M and N represent the number of classes in training and test datasets, respectively. We used a K-fold validation technique to assess unbiased accuracy values for each experiment and compared them with the significance threshold obtained via the permutation test [34]. During the K-fold process, a dataset is divided into K equal parts, and each group is used as a hold-out while the rest is used to train the model. In this work, we divided our data into five groups, which is a standard approach in the field of machine learning.

To assess a time-resolved effect, we aggregated predictions for each 10 min time window and computed a fraction of predictions for all drugs.

In order to verify our ANN-based pattern recognition approach and to be able to directly compare our results with the results of statistical analyses of fish movement macro-parameters, we chose to replicate the experimental setting described in [35]. Using video recordings from the in vivo experiment, we applied our neural network and compared its predictions with the statistical analysis results. We also performed the same calculations described in [35] for consistency.

The model and analysis code were written in Python language (3.8.3) using Pytorch (1.6.1), NumPy (1.19.1), SciPy (1.52.1), Matplotlib (3.5.2), and NetworkX (2.5) libraries.

### 2.7. Time-Resolved Analysis

To assess the timeline of behavioral effects of ethanol, we split the predictions into 10 min time bins and aggregated a fraction of predictions for each class. We compared those fractions with the baseline level obtained via permutation test in each time bin and excluded all non-significant classes from the analysis.

To estimate the significance of time-related changes in the locomotor patterns, we computed Spearman’s correlation coefficient between time and an average predicted concentration in a time bin and applied breakpoint detection to find non-linear changes.

## 3. Results

### 3.1. Concentration-Dependent Effect of Ethanol

We investigated model performance in two independent experiments. We chose this approach because data collected from two independent in vivo experiments tend to be more variable between experiments, so we had to check how robust our approach is when dealing with data that may be different in initial experimental parameters.

The predictions were aggregated into a confusion matrix (Figure 2) and subsequently converted into a network, and node communities were detected with a Louvain algorithm similar to [36].

Predictions are consistent across different K-folds, and three clusters were detected with the following concentrations: 0.5–1%, 1.5–2.5%, and 3.5–4% (Figure 3), with the sole exception of 3.0%. A low concentration of ethanol (0.5%) shows a negligible effect. The model often confuses it with the absence of any drug at all (control recordings), but starting from 1%, the model does not mistake any recording for the control class.

High concentrations of ethanol (3.5–4%) cause significant mortality with rare occurrences of unnatural locomotion (see Table 3). Meanwhile, moderate concentrations (1.5–3%) show the most prominent effect, which is not confused with other concentrations and can be divided into two subclusters: 1.5%, for which accuracy is remarkably high (92%), and 2.0% together with 2.5%, which have a similar effect and are often mutually misclassified (Figure 3).

Interestingly, 3% ethanol shows an edge-case effect and does not belong to any cluster (Figure 3). A considerable fraction of corresponding recordings belong to the cluster with moderate effect, and another part belongs to the cluster with high mortality. It shows that 3.0% is the maximum concentration of ethanol to keep a fish alive and cause significant locomotor activity changes, which is comparable to other studies [37,38]. Interestingly, 3% ethanol shows an edge-case effect and does not belong to any cluster (Figure 3). A considerable fraction of corresponding recordings belong to the cluster with moderate effect, and another part belong to the cluster with high mortality. It shows that 3.0% is the maximum concentration of ethanol, which is comparable to other studies [37,38].

### 3.2. Long-Lasting Effect of Ethanol

We investigated the change in the effect of ethanol for each concentration during the experiment. We found that the behavioral effect varies between different concentrations and can be divided into three behavioral clusters: low concentrations (0.5–1%), medium concentrations (2–3%), and high concentrations (3–4%). In addition, the effect of sobering—where locomotor behavior becomes more similar to that of lower concentrations—shows up in all clusters (Figure 4).

The fraction of predictions with high concentrations drops gradually with time in the low-concentration cluster (0.5–1%), which indicates an effect of sobering. Recordings with a concentration of 1% were sometimes classified as 2% at the beginning. Closer to the end of recordings, most of the predictions belong to either the control class or 0.5%. Recordings for the concentration of 1.5% show the same pattern and the fraction of correct predictions grows with time. (Figure 4, concentrations of 1.0%, 1.5%, and 2.0%).

In the moderate-concentration cluster, the sobering is non-linear, with a huge fraction of high-level concentrations at the beginning and a sharp drop around 30 min after the drug was administered. For instance, recordings of 2.5% ethanol have a high ratio of 3.0% and 3.5% predictions before 30 min, but they are almost absent afterward (Figure 4, concentrations of 2.5% and 3.0%).

In the high-mortality cluster, most predictions belong to the highest (4.0%) concentration of ethanol except at the very beginning of the recordings for 3.5%. We assume this is because of the delayed mortality effect of the 3.5% ethanol. At the beginning of a recording, a fish is still alive but dies soon (in around 10 min) (Figure 4, concentrations of 3.5% and 4.0%).

### 3.3. Non-Linear Nature of Locomotor Patterns

The presence of a drug effect is often assessed by comparing simple locomotor statistics such as speed, velocity, and pause rate [39]. Those metrics are statistics calculated on the animal locomotor data. As with any statistics, some information is lost in the process. This information loss leaves some crucial locomotor patterns undifferentiated/unrecognized. An average speed could be the same, but the shape of the animal tracks could be very different in the two segments.

We analyzed model embeddings—the output of the layer underlying the decision layer with embedded visual features of a locomotor track. We applied Principal Component Analysis (PCA) to such embeddings and found that three components explain more than 90% of the variance in the data.

We analyzed the correlation of samples in the embedding space with the independently computed average speed in a segment and concentration of a drug (the control subgroup was excluded). We found that similar concentrations are close to each other in the embedding space (Figure 5b–d). In addition, samples from the high-concentration cluster (ethanol concentrations of 3.5% and 4.0%) formed a separate cloud.

The average speed of an animal is often used to characterize locomotor behavior. We checked the correlation of the average speed with the first three PCA components. We found that the correlation is not linear, but it creates a gradient—a directed change in speed—which is the most prominent in the first component (Figure 5b).

To compare the predictive power of a model with basic statistics, we calculated the effect size for a difference in speed for all possible pairs of concentrations (Figure 5) and transformed it into a graph. Later, the graph was compared with drug similarity networks obtained from confusion matrices using their modularity values. We found that, despite similar clusters, the effect size cannot distinguish control from low concentrations (Cohen’s d < 0.5, Figure 6) and that the confusion matrix had higher modularity and, therefore, yielded better separation between classes.

## 4. Discussion

Here, we show how an ANN-based pattern recognition approach can be used to examine the change in zebrafish behavior under the influence of ethanol over time. Previously, we knew that fish sober up over time [40], but we could not say exactly when the transition occurred for specific concentrations. The present study shows that fish sober up over time with specific timestamps for each tested concentration. This means that individuals from the group with a higher concentration of ethanol begin to behave like individuals with a lower concentration at the beginning of the observation.

Individuals from the 0.5% group begin to show locomotor activity corresponding to the control group starting from the 60th minute. The locomotor activity of individuals from the 2% group after the 80th minute is classified as the activity of individuals from the 1.5% group. The machine learning approach allowed us to show the biphasic effect of ethanol (Figure 2). However, the use of machine learning also made it possible to identify two additional subclusters with low and medium concentrations (0–1% and 1.5–2.5%), which was not shown by the classical statistical methods using the speed metric (only 2 main clusters were identified, see Figure 5).

The increased duration of the experiment, as compared with visually analyzed setups, helps determine the actual timeframe during which a particular substance concentration is causing an observable effect (Figure 4). The method presented here could be applied to the data with long fish exposure time (90 min in the current study) as well as to the short time of exposure (e.g., standard 5–10 min). The ANN-based classification method itself can help researchers to identify similarities and differences between studied compounds. The temporal analysis could highlight when such similarities occur and how long they last.

Thus, by combining the methods of temporal analysis and clustering, we can compare different concentrations that are similar in their action and evaluate the duration of the effect at a specific concentration. For example, if a slight increase in concentration does not drastically change the behavior pattern (no side effects occur), but prolongs the action of the drug in a non-linear manner, then we can likely increase the concentration to prolong the desired effect. Based on these observations, we can say that using machine learning methods for the classification and temporal analysis of animal behavior is a very promising approach which can be applied to large existing and future datasets from experiments with different concentrations of various substances.

Data extraction in high-throughput screening in zebrafish is a growing field and several studies has already been performed [41,42]. Despite having success with identifying differences in basic locomotor behavior descriptors such as speed or fold change [43] and despite the basic features being enriched with throughput statistical analysis [44,45], it cannot explain complex patterns of activity and is still limited to basic behavior. Some of the other methods concentrate on exploring the locomotor repertoire [46], but they are limited to exploratory analyses and do not support confirmatory analysis.

The presented ML classification approach and the method of temporal analysis of locomotor activity have their own limitations. The number of experiments, the duration of each one, and the number of individuals play a significant role in the applicability of this approach. In the small number of experiments with subthreshold concentrations, it is impossible to reliably state that the classification is based on the substance and not the phenotype of a particular individual. Experiments with short duration may not show differences between drug concentrations, or may not show significant changes in locomotor activity over time. However, for well-studied substances, whose action is noticeably manifested over short periods of time, this method is still suitable with an appropriate selection of time window parameters. Other methods do not have the ability to perform segment-to-segment analysis and can only handle whole exposure time analysis. The presented method is unbiased and can perform temporal analysis with only two coordinates as input data.

The method described in the paper was not yet used to search for patterns and classification in the case of the social behavior of zebrafish individuals. To continue advancing the method, we want to apply it to the analysis and classification of social behaviors and enhance its performance for analyzing experiments with substances that have a short-lasting effect.

## Figures and Tables

**Figure 1 biomedicines-11-03215-f001:**
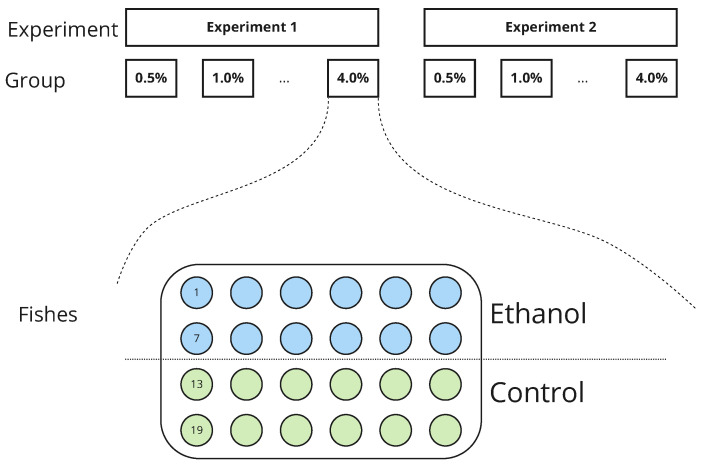
Schematic representation of two ethanol experiments. In each experiment, zebrafish larvae were exposed to an ethanol solution with different concentrations of the drug ranging from 0% (control group) to 4%. Animals were randomly divided into control and ethanol subgroups of equal size (N = 12 in each subgroup). Zebrafish were placed into a round tank and their movements were recorded during 90 min long sessions.

**Figure 2 biomedicines-11-03215-f002:**
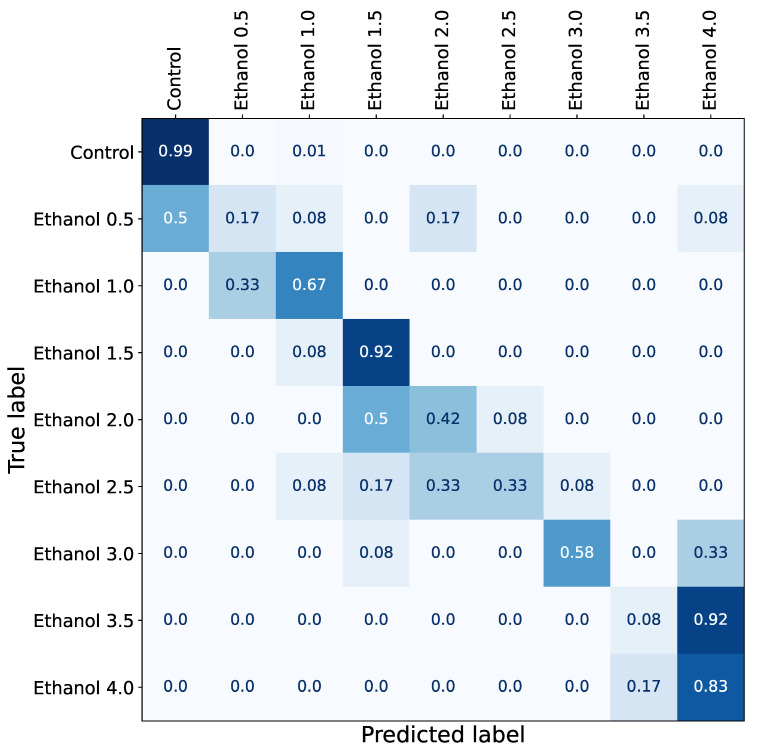
Graphic summary of prediction accuracy data generated in the first computational experiment. In this experiment, we trained a neural network model using data from different concentrations of ethanol ranging from 0% (control group) to 4% and aggregated its predictions into a confusion matrix to assess the similarity between drugs. Low-level concentrations (0.5 and 1.0), medium-level concentrations (2.0 and 2.5%), and high-level concentrations (3.5 and 4.0) show similar behavioral changes, while 1.5% and 3.0% have clear effects. In addition, almost all concentrations are different from the control group apart from the lowest concentration (0.5%), which can be mixed with control in half of the samples.

**Figure 3 biomedicines-11-03215-f003:**
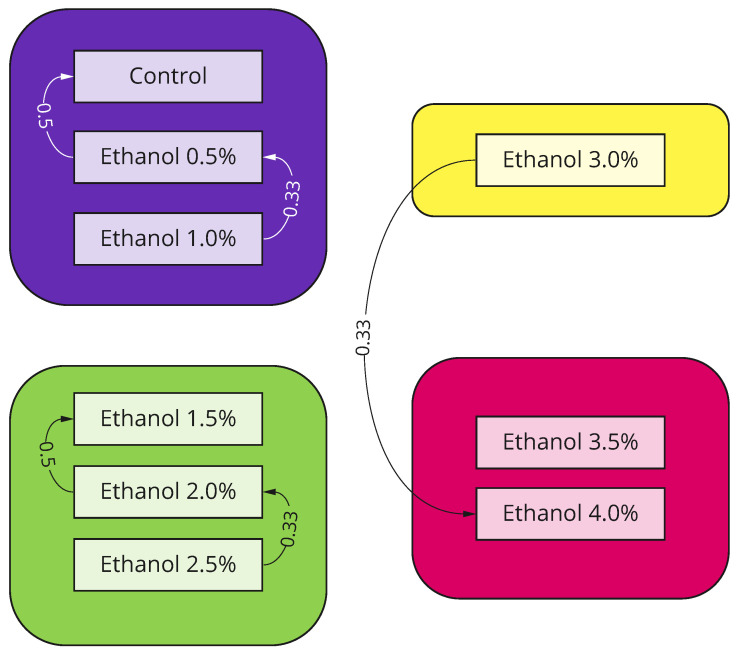
To assess clusters of locomotor behavior, we applied a Louvain clustering algorithm to the confusion matrix data obtained in a computational experiment. In total, four communities were detected: Low-level concentrations (0%, 0.5%, and 1.0%, purple) were merged into a cluster with almost negligible locomotor changes, which are often confused with the control group. Medium-level concentrations (1.5%, 2.0%, and 2.5%, green) were merged into a cluster with a noticeable effect that is significantly different from the control data. High-level concentrations (3.5% and 4.0%, red) recordings were merged into a separate cluster with no connections to other clusters except the concentration of 3.0%.

**Figure 4 biomedicines-11-03215-f004:**
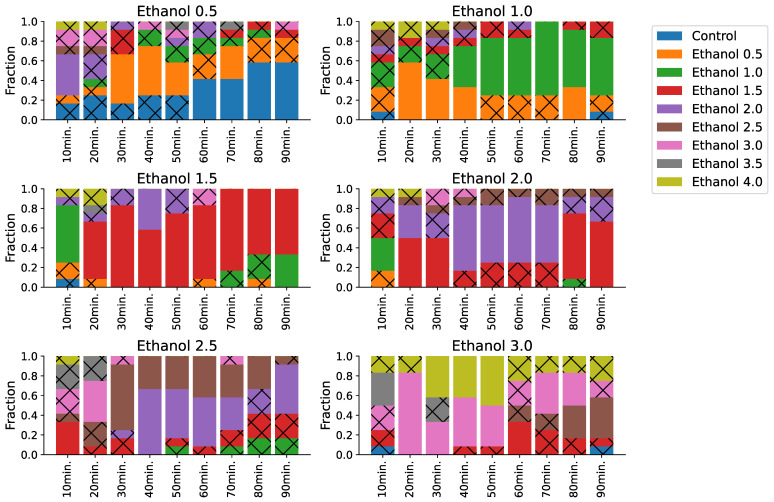
Timeline of ethanol concentration predictions. For each true class, we aggregated corresponding predictions and computed a fraction of predicted labels in each 10 min bin, a bar plot indicating the fraction of those predictions. We excluded samples that belong to the control and high-concentration (3.5%, 4.0%) classes because their predictions were almost/nearly constant, without noticeable time-related effects. Hatched sections indicate non-significant predictions. A concentration of 0.5% shows an effect duration of about 60 h and becomes similar to the control afterward. Medium concentrations (1.0%, 1.5%, and 2.0%) show a prolonged effect that does not drop significantly for up to 90 min. High-level concentrations (2.5% and 3.0%) show a strong effect at the start and up to 30 min and are often confused with higher concentrations, but they become weaker with time.

**Figure 5 biomedicines-11-03215-f005:**
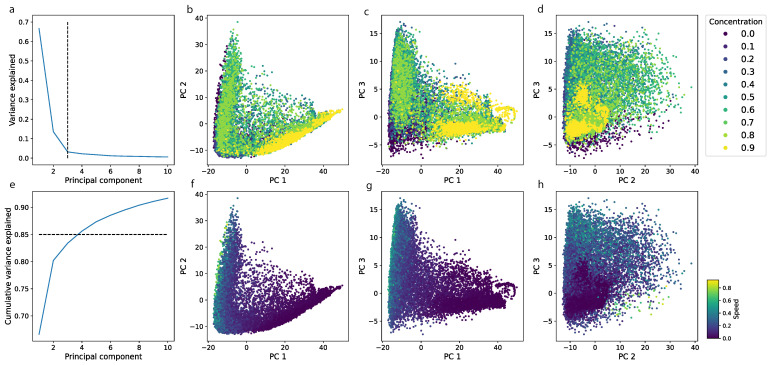
In order to assess the possible correlation between high-order features extracted by a neural network and basic features such as speed, we applied a PCA to the output of the one-by-last layer of the neural network. We analyzed the first three principal components because they explain more than 85% of the variance in the data (proportion of variance explained is shown in (**a**), cumulative variance explained is shown in (**e**)). (**b**,**f**) show the scatter plot of the first two components, (**c**,**g**) for the first and third components, and (**d**,**h**) for the second and third components. Samples with the same concentration have similar representations in the embedding space and are grouped into clouds (**b**,**c**) and two functional communities, low and high concentrations). We also found that the average speed during a segment plays a role and forms a gradient alongside the first three components, which is the most prominent in the first and third (**d**,**h**).

**Figure 6 biomedicines-11-03215-f006:**
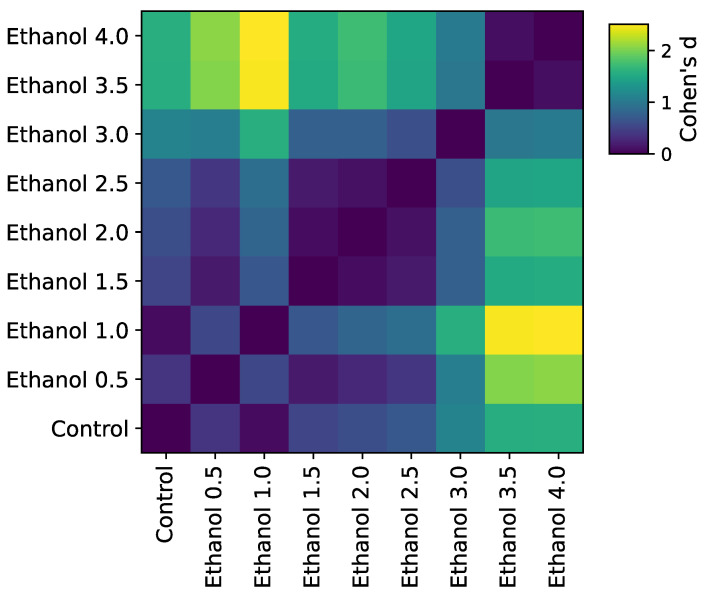
Pairwise Cohen’s values for an average animal speed in a track segment. In contrast to pairwise classification similarity (Figure 2), classical features cannot separate different concentrations with high confidence and merge them into a single cluster (e.g., all medium-level concentrations, 1.5–3.0%, have similar average speeds and, therefore, belong to the same behavioral community).

**Table 1 biomedicines-11-03215-t001:** Zebrafish age in experiments.

	Experiment 1	Experiment 2
Groups 0.5–3%	11 dpf	12 dpf
Groups 3.5–4%	12 dpf	13 dpf

**Table 2 biomedicines-11-03215-t002:** Chemical substances for a final ethanol concentration. The 4% solution obtained from the first dilution (47.5 mL in total) was used for further dilution of the remaining concentrations (31 mL for the other concentrations). The volume of the solution obtained for each concentration was sufficient to fill 12 wells of one plate, with 0.5 mL per well.

Ethanol Solution	Neutral Solution	Final Solution
4% 1 mL	7 mL	0.5%
4% 3 mL	9 mL	1.0%
4% 5 mL	8.3 mL	1.5%
4% 3 mL	3 mL	2.0%
4% 5 mL	3 mL	2.5%
4% 7 mL	2.3 mL	3.0%
4% 7 mL	1 mL	3.5%
95% 2 mL	45.5 mL	4.0%

**Table 3 biomedicines-11-03215-t003:** Fraction of dead fish per each solution.

Solution	Fraction of Dead (Non-Moving) Fish
Control	0%
Ethanol 0.5–3.0%	0%
Ethanol 3.5%	9%

## Data Availability

Data are contained within the article.

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
