# Peer review of "Artificial Neural Network (ANN)-Based Pattern Recognition Approach Illustrates a Biphasic Behavioral Effect of Ethanol in Zebrafish: A High-Throughput Method for Animal Locomotor Analysis"

_biomedicines, 2023, doi:10.3390/biomedicines11123215_

Round 1
Reviewer 1 Report
Comments and Suggestions for Authors
The authors provide a novel study describing an interesting and new strategy to evaluate behavioural outcomes of zebrafish when exposed for longer periods during early developmental stages. Overall, this is a proof-of-concept with the authors presenting a well written and scientifically valid method that can be applied by others. Yet, there are still some minor changes that need to be performed:
- Include a conclusion of the work in the abstract.
- Avoid the use of abbreviation in the keywords.
- L23, remove the dot after the references.
- L80, it is stated that 25 larvae per 200 mL egg water were used. Yet, L75 refers a total of 384 larvae. How were these larvae distributed in the groups?
- L86, review table 2 as 45.5 ml does not fit a well of a 24-well plate.
- L93, review the n described as it is referred that one single larva per well was used and half a plate was used (n=12 not 15).
- L105, “dosage” should be changed to concentration. Review along the manuscript.
Author Response
- Include a conclusion of the work in the abstract.
We added a conclusion of the work in the abstract.
- Avoid the use of abbreviation in the keywords.
We replaced the ANN abbreviation with its complete version.
- L23, remove the dot after the references.
We removed the dot.
- L80, it is stated that 25 larvae per 200 mL egg water were used. Yet, L75 refers a total of 384 larvae. How were these larvae distributed in the groups?
We added some clarification in section 2.1. 384 larvae were used for the experiment, but the number of previously outbred eggs were higher.
- L86, review table 2 as 45.5 ml does not fit a well of a 24-well plate.
We added clarification for Table 2. 45.5 ml is a neutral solution, and the total amount of 4% solution is 47.5. Then it was used for all further dilutions for all other concentrations. All represented in the Table 2 concentrations are 6ml or more, which is fitting in 12 wells with 0.5 ml in each.
- L93, review the n described as it is referred that one single larva per well was used and half a plate was used (n=12 not 15).
We reviewed the number and corrected the text accordingly.
- L105, “dosage” should be changed to concentration. Review along the manuscript.
We replaced dosage with the concentration across the manuscript text.
Reviewer 2 Report
Comments and Suggestions for Authors
The paper is not bad but it is not good either. I have however some comments: in introduction please cite also original papers but not only reviews. There is a lot of good papers which showed discovery of new compunds with the aid of zebrafish (eg.epilepsy, schizophrenia etc). Keywords: dont use ANN abbr in keywords.
What does it mean „local suppliers”? Please specify, also provide information about ZF background. Number of ethical permission etc missing, please provide. Also describe this part in details step by step. Description of methodology is very weak.
Why authors analyzed effect of ethanol in such „late” period of developemnt? Usually people analyze it at 5 dpf there is no need ethical permission. Discuss it, compare with exisiting data. Make some table with yr data and other papers. I ve not learnt much from this paper to be honest. Discussion is very week. Authors did not convince me their method is better than other methods.
Comments on the Quality of English LanguageIts ok
Author Response
- In introduction please cite also original papers but not only reviews.
We reviewed the introduction and cited papers with original research about the usage of zebrafish in various biomedical research.
- Keywords: dont use ANN abbr in keywords.
We replaced the ANN abbreviation with its complete version.
- What does it mean „local suppliers”? Please specify, also provide information about ZF background. Number of ethical permission etc missing, please provide. Also describe this part in details step by step. Description of methodology is very weak. Why authors analyzed effect of ethanol in such „late” period of developemnt?
We added the explanation of using adult zebrafish models to the ‘Animals and housing’ section. Discuss it, compare with exisiting data. Make some table with yr data and other papers.
We rewrote sections 2.1 and 2.2 and added information about ZF, its background, housing conditions, information about used compounds, ethical permissions, methodology, and reasons for the usage of larvae with late-stage development.
- Authors did not convince me their method is better than other methods.
We reviewed the manuscript and added a comparison with other methods in the discussion.
Round 2
Reviewer 2 Report
Comments and Suggestions for Authors
More details about strain of fish needed.
Provide nr of ethical permission for experiments
Author Response
- More details about strain of fish needed.
We revised the text in the 'Animal and Housing' section according to the response from the laboratory where the experiments were conducted.
- Provide nr of ethical permission for experiments
We added an Ethical Statement section, providing the number of the research protocol approved by the ethics committee.